# Normal-Incidence Germanium Photodetectors Integrated with Polymer Microlenses for Optical Fiber Communication Applications

**DOI:** 10.3390/s24134221

**Published:** 2024-06-29

**Authors:** Yu-Hsuan Liu, Chia-Peng Lin, Po-Wei Chen, Chia-Tai Tsao, Chun-Chi Lin, Tsung-Ting Wu, Likarn Wang, Neil Na

**Affiliations:** 1Institute of Photonics Technologies, National Tsing Hua University, Hsinchu 30013, Taiwan; rich22368520@gmail.com (Y.-H.L.); lkwang@ee.nthu.edu.tw (L.W.); 2Artilux Inc., Hsinchu 30288, Taiwan

**Keywords:** germanium photodiode, polymer microlens, photon-acid diffusion, reliability test, high speed, optical communication

## Abstract

We present a novel photon-acid diffusion method to integrate polymer microlenses (MLs) on a four-channel, high-speed photo-receiver consisting of normal-incidence germanium (Ge) p-i-n photodiodes (PDs) fabricated on a 200 mm Si substrate. For a 29 µm diameter PD capped with a 54 µm diameter ML, its dark current, responsivity, 3 dB bandwidth (BW), and effective aperture size at −3 V bias and 850 nm wavelength are measured to be 138 nA, 0.6 A/W, 21.4 GHz, and 54 µm, respectively. The enlarged aperture size significantly decouples the tradeoff between aperture size and BW and enhances the optical fiber misalignment tolerance from ±5 µm to ±15 µm to ease the module packaging precision. The sensitivity of the photo-receiver is measured to be −9.2 dBm at 25.78 Gb/s with a bit error rate of 10^−12^ using non-return-to-zero (NRZ) transmission. Reliability tests are performed, and the results show that the fabricated Ge PDs integrated with polymer MLs pass the GR-468 reliability assurance standard. The demonstrated photo-receiver, a first of its kind to the best of our knowledge, features decent performance, high yield, high throughput, low cost, and compatibility with complementary metal-oxide-semiconductor (CMOS) fabrication processes, and may be further applied to 400 Gb/s pulse-amplitude modulation four-level (PAM4) communication.

## 1. Introduction

The use of Ge for the design and fabrication of photo-receivers for high-speed optical fiber communication has become prevalent in recent years. Due to its broadband absorption range from roughly 300 nm to 1600 nm wavelengths, widespread adoption of Ge for short-range multi-mode fiber (MMF) communication at 850 nm and mid-to-long range single-mode fiber (SMF) communication at 1310 nm and 1550 nm have been achieved. Moreover, PDs made from Ge are intrinsically CMOS fabrication process compatible, enabling low-cost solution for mass production. In the literature, remarkable progress on Ge PDs with low dark current, high responsivity, and high bandwidth, have been realized in both waveguide and normal-incidence geometries [1,2,3,4,5,6,7,8,9,10,11,12,13,14]. However, when applying Ge PDs to photo-receiver modules, efficient optical coupling remains a challenging topic. For waveguide geometry, optical couplers such as grating coupler or inverted taper coupler are usually needed, which introduce additional fiber-to-coupler insertion loss and a stringent alignment requirement [14,15,16,17]. For normal-incidence geometry, the trade-off between bandwidth, determined by RC time constant and transit time constant, and external responsivity, determined by PD size and PD thickness, impose a complication in optimizing PD design parameters [8,14]. With the demand for increasingly higher data rates over time, the size of a PD needs to be shrunk to reduce the capacitance and therefore attain the desired receiver bandwidth, consequently reducing the optical aperture size of the PD. Therefore, it is becoming harder and harder to achieve low coupling loss and high alignment tolerance when the photo-receiver is packaged in an optical module.

One approach to increase the optical aperture size is to incorporate a microlens (ML), in which parameters such as radius of curvature, focal length, contact angle, and refractive index should be carefully designed to derive a high optical performance [18]. At the same time, when fabricating the ML, one needs to consider (1) materials that are reliable and possess proper contraction coefficient and viscosity, and (2) fabrication processes that are efficient and low cost. Depending on the applications, choosing suitable materials and fabrication processes to develop a high-performing ML can be a challenging topic. There have been numerous works demonstrating various approaches including reflow [19], ink-jet [20], imprint [21], and grayscale lithography [22], to name a few. In the reflow method, a layer of material is coated on the substrate and patterned via optical lithography to form a flat-top pillar. The pillar is then heated to a specific temperature for well-controlled time duration to form the reflow lens. Note that this method has been applied in mass production of MLs for sensors and imagers. In the ink-jet method, the material is placed in a pipe and then dispensed from a dropper onto the substrate. The size and shape of the lens are dependent upon the volume of droplets and the contact angle between the material and the substrate. Note that for the ink-jet method, highly accurate and stable-yet-fast dispensing tools are needed to overcome the issues of poor alignment and low throughput. In the imprint method, the lenses are formed using a mold as the master replica. Note that for this method, how to fabricate a precise mold and how to de-mold after the imprint are crucial. In the grayscale lithography method, direct patterning the photoresist with a grayscale mask to match the lens shape is first applied. Then the photoresist pattern is transferred to the layer below the photoresist through dry etching. Note that while this method can be handled by a mass production facility in a straightforward manner, the cost of gray-scale mask is high, and the dry etch process would cause roughness over the ML surface, reducing its optical quality due to light scattering.

In this work, we demonstrate an optimal method to fabricate polymer-based ML through photon-acid diffusion [23]. We experimentally show that MLs with an effective optical window size as large as 54 µm can be mass produced and pass the related reliability qualifications. Such an ML allows the maintenance of high external responsivity through low coupling loss and high alignment tolerance when interfacing a PD. Moreover, we demonstrate the integration of a Ge PD with the fabricated large polymer ML. The optical aperture size is thus effectively enlarged and the trade-off between PD bandwidth and PD size becomes less stringent. The optical properties of the polymer ML and the Ge PD are designed through running a commercial finite-difference time domain (FDTD) simulation software. The designed device, i.e., a Ge PD integrated with a polymer ML, is then fabricated and duplicated to form a one-dimensional (1D) array on 8-inch Si wafers. The dark current, responsivity, 3 dB bandwidth, and effective aperture size of the integrated device are characterized through either wafer-level or die-level electro-optical testing. Based on these results, selected devices are packaged and are subjected to an eye diagram as well as sensitivity measurements. In addition, reliability tests based on GR-468 reliability assurance standard are also performed. With the demonstrated Ge PD integrated with polymer ML, 100 Gb/s NRZ systems can be realized by four-channel 25 Gb/s devices having a 3 dB bandwidth of >17.5 GHz, and 400 Gb/s PAM4 systems can be likewise realized by four-channel 50 Gb/s devices with a 3 dB bandwidth of >35 GHz in the future [24,25,26].

## 2. Polymer ML Development

### 2.1. Polymer ML Design and Simulation

The relation between ML dimensions and its focal length can be found in Ref. [18] and is re-stated as follows. Suppose the profile of an axially symmetrical plano-convex ML is considered and expressed by a function *h*(*r*), as shown in Figure 1a. When such a profile is spherical, it can then be described as follows:(1)$$h(r)=\frac{1}{R}\frac{r^{2}}{1+\sqrt{1-r^{2}/R^{2}}}$$

Note that *h*(*r*) represents the height of the ML’s convex surface as a function of the position *r*, and *R* is the radius of curvature of the sphere. Assuming the refractive index of the lens considering material, dispersion is represented *n*(*λ*), the focal length *f* can then be expressed as follows:(2)$$f=\frac{R}{n(\lambda )-1}$$
and the contact angle *α* at the lens border can be expressed as follows:(3)$$\sin\alpha =\frac{r_{0}}{R}=\frac{r_{0}}{f(n(\lambda )-1)}$$
where *r*_0_ represents the size of the lens. Here, we consider a 50/125 µm MMF at 850 nm connecting to a custom Rx lens module that focuses the MMF output light down to 30 µm in size to serve as the input light to the ML. The high-order mode profile of the MMF in the simulation is chosen so that the numerical aperture of the input light to the ML matches the experimental data. The FDTD simulation result is shown in Figure 1b, in which the focal length of the ML is set to be 45 µm in the buffer layer. The refractive index of the polymer used for both the ML and the buffer layer is set to be 1.5. Up to 99% of the optical power is focused inside a 24.5 µm optical window at the focal plane, demonstrating a good performance of multimode focusing through the ML. This serves as the baseline design of the 25 µm optical aperture size used for the Ge PD to be demonstrated in the following sections.

### 2.2. Fabrication of Polymer MLs

There are a wide varieties of polymer materials available for fabricating MLs. The properties of these materials that should be paid special attention to are refractive index, viscosity, contraction coefficient, and process temperatures. The refractive indices of most polymer-based chemicals are comparable to that of glass, about 1.5. The viscosity depends on the material itself and the interfacial bonding with the substrate. Due to the relatively large contraction coefficients, the volume of commercial polymer materials may shrink by <5% after dehydration. The process temperature for polymer ranges from room temperature to <400 °C, and one should select a proper temperature not to degrade the Ge PD beneath the polymer ML. With the additional consideration of passing the reliability qualifications for optical fiber communication applications, three commercially available polymers, AZ [27], ORMO [28], and SU8 [29], are chosen for our ML development (the material properties can be found in Refs. [28,29,30]. E.g., the viscosity of the SU8 is 4400 cSt.). To determine the most suitable polymer for the ML, we spin coat a thin film on a glass substrate and then heat the substrate to 125 °C. The transmission spectra derived from UV-VIS (ultraviolet–visible) spectroscopy of ORMO and SU8 after the materials being baked at 125 °C for different time durations are shown in Figure 2. Note that AZ is not included in Figure 2 because it does not survive the 125 °C baking process where cracks appear after 120 h. ORMO can sustain nearly 100% transmission for wavelengths longer than 700 nm after 800 h, while SU8 shows a less than 2% drop in transmission for wavelengths longer than 850 nm after 800 h. The transmission drop is due to the chemical changes in the polymer material responding to the thermal budgets. At the visible wavelengths, for both ORMO and SU8, the transmissions drop rapidly with baking duration. However, because the target operation wavelength is 850 nm, the material decay at shorter wavelengths can be overlooked, both ORMO and SU8 are deemed viable for our application. Finally, after further studies, ORMO is excluded from the consideration due to its high viscosity, leaving SU8 as the only logical choice. The extensive use of SU8 in the industry for its durability and reliability is another reason that we apply SU8 in our work.

In the following, two types of ML process developments base on photon-acid diffusion with SU8 are discussed. The process flow of the Method 1 formation process is shown in Figure 3: a 20 µm thick SU8 layer is spin coated on a substrate at 3000 rpm for 40 s, and then baked at 150 °C for 5 min. Then, the 1st photolithography is applied to define the pedestal region with a UV dosage of 140 mJ/mm^2^. A 30 µm thick SU8 layer is then spin coated over the 1st SU8 layer at 2000 rpm speed for 40 s, followed by a soft bake at 120 °C for 5 min. After a process of edge bead removal (EBR), the 2nd photolithography is applied to generate photon-acid in the 2nd SU8 layer at the ML region with a UV dosage of 100 mJ/mm^2^. A thin SU8 layer is then spin coated over the 2nd SU8 layer at 1000 rpm for 40 s, followed by no soft bake, and placed a in clean dry air (CDA) purging chamber for an hour for photon-acid diffusion. Within this hour, the photon-acids diffuse outward uniformly to form a concentration gradient. Post exposure baking (PEB) at 70 °C for 5 min and follow-up development remove the additional polymer without a crosslink, forming the shape of the lens. The following is hard baked at 150 °C for 5 min, yielding a spherical SU8 ML on a pedestal with a diameter of 54 µm. The main drawback of this process is the formation of a distorted spherical ML due to the two exposures in the experiment.

To circumvent the problem of the distorted spherical ML resulting from the Method 1 formation process, Method 2 formation process is proposed and studied. The process flow is shown in Figure 4, which is different from the process flow in Figure 3 by using only one exposure when there are two SU8 layers, instead of one exposures where one is used for the 1st SU8 layer, and the other is used for both the 1st and the 2nd SU8 layers. The Method 2 formation process not only solves the problem of distorted spherical MLs, but also simplifies the fabrication process, making it possible to achieve mass production with high throughput and low cost. It should be noted that both methods originate from the same design concept: the lithography defines the ML and the pedestal dimensions, and the UV dosage controls the photon-acid, in which a sufficiently long idle time is needed for the photon-acids to diffuse uniformly. With properly chosen thicknesses for both the ML and the pedestal, a giant ML with good spherical lens shape can be achieved. The ML performance and dimensions are examined next.

### 2.3. Experimental Results of Polymer ML

The optical microscope (OM) images in Figure 5 show two MLs on pedestals fabricated using Method 1 with different dosages at the 2nd exposure step. Note that the two MLs on pedestals are placed at the opposite side of a spacer, prepared for protecting the devices from damages during the stage of wafer dicing and chip packaging. The pedestal is built from the 1st SU8 layer, which was formed by the 1st spin-coating and the 1st exposure. After the 2nd SU8 layer is coated, we have two essentially non-identical materials, i.e., the 1st exposed layer and the 2nd non-exposed layer, even though they originated from the same polymer material. The interface between them is hereafter referred to as “inhomogeneous interface”. Since the 2nd SU8 layer without UV exposure does not dissolve well in the 1st SU8 with UV exposure, the correspondingly weak adhesion through the inhomogeneous interface as well as the resultant anisotropic photo-acid diffusion lead to a distorted spherical ML with its vertex shifted away from the center. After adjusting the UV dosage to balance the photon-acid generation within the 1st and 2nd SU8 layers, the best spherical ML is achieved at the UV dosage of 100 mJ/mm^2^, whereas its shape becomes worse with increasing UV dosages. Still, Method 1 cannot produce a satisfactory ML shape when its diameter is roughly larger than 45 µm.

On the other hand, an ML on a pedestal fabricated by Method 2 is shown in Figure 6. Since the 2nd SU8 layer is processed right after the 1st SU8 layer so that both layers are exposed to UV at the same time, the dimensions of the ML and the pedestal are defined simultaneously through the generated photon-acids. This way, one can avoid the problem of inhomogeneous interface observed in Method 1. In Figure 6a, the scanning electron microscope (SEM) of a nearly perfect spherical ML on a pedestal with a diameter of 54 µm is shown. The total height between the vertex of the ML and the bottom of the pedestal is about 65 μm, which is close to the design target shown in Figure 1b. In Figure 6b, the SEM image of arrays of ML separated by arrays of spacers is shown. High uniformity of the fabricated MLs on pedestals using Method 2 can be observed. Note that the height of the spacer is intentionally made higher than that of the ML on pedestal so that the spacers can protect the surface of the MLs on pedestals from scratching when wafer dicing and chip packaging are performed. Figure 6c is the 3D profile of the fabricated ML using Method 2 for checking ML sphericity. The result shows clear concentric circular interference patterns, which means a nearly ideal symmetric lens shape is derived. The un-resolved slopes within the dark region are due to the tool limitation, but they can be seen in the SEM result as continuous and smooth. These results demonstrate a promising method for fabricating giant polymer ML arrays on an 8-inch Si substrate with a high yield, low thermal budget, and cost-efficient process.

## 3. Full Device Fabrication

### 3.1. Ge Film Deposition on Si Substrate

In the literature, methods such as molecular beam epitaxy (MBE) [30,31,32], reduced pressure chemical vapor deposition (RPCVD) [33,34,35], rapid melt growth (RMG) [36,37,38], and plasma enhanced chemical vapor deposition (PECVD) [39], have been used to grow high-quality Ge films on Si substrates. Due to the 4.2% lattice mismatch between Ge and Si, defects including misfit dislocation and threading dislocation appear in the Ge film deposited on the Si substrate. These defects may act as photo carrier recombination centers and dark current generation centers [40,41,42], and their density can be calculated by cross-sectional transmission electron microscope (TEM) and the etch pit method [43].

In this study, we applied RPCVD to grow a 1 µm thick Ge film on a Si substrate (100 plane). Figure 7a shows the cross-sectional TEM image of the 1 µm thick Ge film deposited on a Si substrate. Misfit dislocations at the Ge/Si interface along with several threading dislocations can be observed. In the Ge region away from the interface, most defects merge and disappear, resulting in a relaxed Ge layer with a high degree of single crystalline structure. Figure 7b shows the top-view atomic force microscope (AFM) image, and Figure 7c shows the top-view SEM image, both derived after applying an etch pit method. The total defect density is then estimated to be on the order of 1 × 10^7^ cm^−2^.

### 3.2. Ge PD Fabrication

Ge p-i-n PDs are fabricated with a process compatible with standard CMOS technology. After the 1 µm thick Ge film is grown on an 8-inch p-type Si substrate by RPCVD, the device active area is defined by double mesa patterning with i-line lithography, followed by dry etching processes. The p-i-n junction is formed by first implanting phosphorus into the Si substrate and then implanting boron into the Ge film surface. A film of 600 nm thick silicon dioxide (SiO_2_) is subsequently deposited by PECVD as an inter layer dielectric (ILD), followed via opening with a reactive ion etching (RIE), and Ti/TiN/Al/TiN deposition/patterning with physical vapor deposition (PVD)/transformer-coupled plasma (TCP) etching. A silicon nitride layer (SiN) is then deposited as an anti-reflection coating, followed by removal of the silicon nitride on top of the metal pad. The schematic plot of the device structure is shown in Figure 8a. In Figure 8b,c, the cross-sectional SEM image and top-view OM image of the fabricated devices are shown.

### 3.3. Ge PD Integrated with SU8 ML

The SU8 MLs are directly processed on an 8-inch Ge PD device wafer, which require no extra carrier wafer or molding. During the fabrication, the first step is to spin-coat the 1st SU8 layer on the device wafer, followed by a soft bake. Then, the 2nd SU8 layer is spin-coated on top of the 1st SU8 layer, followed by another soft bake. The next step is to define the targeted ML dimensions using a contact aligner for the process of photolithography. Since SU8 works as a negative photoresist, exposed regions would crosslink through the exposure to UV light. With a proper device design and process tuning in UV dosage, SU8 volume, baking temperature, and so on, the exposed SU8 region can form a half-ball lens shape standing a pedestal. The unexposed SU8 region can be removed by the developer. After the shape of the SU8 ML were confirmed, we moved to the last step, i.e., a hard bake to solidify the SU8 ML. Note that since SU8 is removable before the final hard bake, the Ge PDs beneath the SU8 remain intact if the MLs are removed due to incorrect processing. This advantage, i.e., the SU8 ML process can be re-worked when a process goes wrong before the final hard bake, gives our ML a better competitive edge over MLs fabricated by other approaches. Figure 9a shows the SEM image taken at a 45-degree angle, showing an ML integrated on a Ge PD. Note that the abnormal pattern shown on top of the ML region is caused by the charging effect of the electrical beam, since we would like to derive the actual size of the ML and so the ML is not protected by Pt coverage. The side view of the SU8 ML can be seen in Figure 9b, with a diameter of 54 µm enlarged aperture and a total height of about 65 µm. Figure 9c shows the OM image of four-channel Ge PDs with integrated SU8 MLs. These results confirm that our integration process is promising for mass production on an 8-inch Si substrate.

## 4. Device Characterization

### 4.1. I-V Characteristics and Frequency Response

The I-V characteristics of the fabricated Ge p-i-n PDs with and without integrating the polymer MLs are characterized using the Keysight B1500 semiconductor device analyzer. The dark currents of 29 µm diameter devices without MLs are measured to be ~138 nA at −3 V bias. By collecting the dark currents from devices in different diameters without ML, the bulk dark current density J_bulk_ and the surface dark current density J_surface_ are extracted to be around 9.25 mA/cm^2^ and 7.56 µA/cm, respectively. Further measurements show that there is almost no observable difference between the dark currents measured from the Ge p-i-n PDs with and without the polymer MLs. Finally, the responsivities of the 29 µm diameter devices without MLs are measured using an 850 nm fiber laser connected to a pigtailed focuser. The responsivity is measured to be ~0.6 A/W at −3 V. The typical dark and photo I-V curves measured from the 29 µm diameter devices without MLs are shown in Figure 10a, and the statistics of dark current at −3 V bias from three batches of the fabricated Ge p-i-n PDs are shown in Figure 10b. Each batch contains 600 DUTs in the box plot, and the outliers are from the ugly dies.

To obtain the 3 dB bandwidths of the fabricated Ge p-i-n PDs, we use the Keysight PNA N5244B network analyzer with a working frequency ranging from 10 MHz to 43.5 GHz to measure the S parameters. An iXblue optical modulator is used to modulate the laser light at 850 nm. Figure 11a shows the frequency response of a fabricated 29 µm diameter device at −3 V bias, indicating a 3 dB bandwidth of 22.5 GHz. Figure 11b shows the statistics of 3 dB bandwidth at −3 V bias from three batches of the fabricated Ge p-i-n PDs. Each batch contains 120 DUTs in the box plot, and the outliners are from the ugly dies.

Table 1 summarizes the performance of the vertical Ge p-i-n PD reported in the literature in recent years. It can be seen that in this work that we have demonstrated the largest QE-BW product, and the dark current densities are comparable with some of the best results. Moreover, among the works showing their QE-BW products, we also demonstrated the largest optical aperture to date.

### 4.2. Comparison of Optical Window Size

In order to demonstrate the improvements of the effective optical window size through ML integration, we mount the pigtailed focuser, which is connected to an 850 nm fiber laser, on a high-precision automatic linear stage, and sweep it across the fabricated PDs with MLs. Figure 12 shows the normalized photo response versus horizontal displacement. The blue line represents the PD without ML, and the orange one represents the PD with ML. Note that the displacement is relative between the center of the focuser and the PD. It can be seen that the effective optical window size, defined by the full width at half maximum (FWHM) of the curves in Figure 12, is increased from ~24 µm (when there is no ML) to ~52 µm (when there is an ML), demonstrating the effectiveness of the integrated ML. Note that such sweepings are performed in both x and y directions to examine the symmetry of the integrated ML, and we find no particular difference between the 2 directions implying good circular symmetry of the integrated ML.

### 4.3. Sensitivity of the Receiver Module

A chip-on-board (COB) module including our Ge PDs integrated with polymer ML, commercial transimpedance amplifiers for acquiring voltage signal from the PD, 850 nm vertical-cavity surface-emitting lasers (VCSELs), and drivers for the lasers, are used for the eye diagram and bit error rate measurement. A non-return-zero (NRZ) pseudorandom binary sequence (PRBS) data stream with a length of 2^31^-1 is applied to the driver, and either a digital communication analyzer (Keysight, E8363B) or a signal quality analyzer (Anritsu, Atsugi, Japan, MP1800A) is connected to the COB module for the eye diagram and bit error rate measurement, respectively. In Figure 13a–d, the measured eye diagrams from the output of a four-channel photo-receiver using this COB module with input data rates all set at 25.78 Gbps and device biases all set at −3.3 V are shown, in which clear eye diagrams can be observed. The temperature of the COB module is controlled to be 25 °C. Then, the sensitivities are measured to be −9.2 dBm with a bit error rate (BER) of 10^−12^, is competitive compared to commercial products. Note that because the extinction ratio of the signals is 4.78 dB, the sensitivities defined by optical modulation amplitude (OMA) and average optical power (AOP) are almost the same. 

### 4.4. Passing the Reliability Test Criteria

In the following, reliability tests are performed in accordance with the definitions and requirements in document Telecordia GR-468-CORE for optoelectronic systems, such as high temperature operation (HTOL) test, temperature cycling (TC) test, damp heat (DH), and COB module wire-bond test. All devices under test (DUT) are selected from three consecutively fabricated batches, and they need to pass all these reliability tests to demonstrate consistency and reliability in applying the devices for products.

Figure 14a shows the results of the HTOL test over a total of 78 DUT from the three consecutively fabricated batches. During testing, the dark currents of the Ge p-i-n PDs built on an evaluation board are initially biased at −3 V bias and 25 °C for dark current measurements. Then, these DUT are biased at −6 V bias and 175 °C for 168 h, 500 h, 1000 h, 1500 h, and 2000 h. After each heating duration, the DUT are cooled down back to the room temperature for dark current measurements at −3 V bias. Figure 14a shows the measured dark currents at the six check points. As the failure criterion is set to be a dark current variation exceeding 50% of the initial value, all DUT pass the HTOL test for the heating durations up to 2000 h. The fabricated devices also pass the high temperature storage (HTS) test over a total of 66 DUT from the three consecutively fabricated batches. The HTS test is performed at 135 °C for a duration of 2000 h, as shown in Figure 14b, in which the variations of QE measured after 2000 h are shown. The failure criterion for the HTS test is set to be a QE variation exceeding 5% of the initial value after 2000 h. The two testing results reveal that the SU8 polymer material is strong enough to withstand thermal stress, implying long product lifetimes. In the TC test, a total of 75 DUT, subjected to 500 cycles of −40 °C/80 °C at 10 °C change per minute temperature variation rate, all pass the same criterion as in HTOL test. In the DH test, a total of 51 DUT, subjected to 85 °C temperature and 85% relative humidity for 500 h, all pass the same criterion as in HTOL test. The COB module wire-bond test including ball pull force and shear force, which indicates the wire-bond strength between a device pad and a pad on the evaluation, results in a strength of 9.19 gf and 13.78 gf, respectively, both conforming to the success criteria. Finally, the electrostatic discharge (ESD) test, which evaluates the electrical pulses that may occur during probing, wire-bonding, pick-and-place, and so on, is performed. The most common ESD test is the Human Body Mode (HBM) test, and results from the ESD test under such a mode show that our devices could withstand electrical pulses at a forward bias of 2000 V and a reverse bias of 250 V. These excellent reliability test results showcase that our devices can be used in commercial products.

## 5. Conclusions

For the first time, we demonstrate Ge p-i-n PDs integrated with polymer MLs for an optical communication receiver at 850 nm wavelength through an optimal method of photon-acid diffusion. The unique property of this device, i.e., it is capable of absorbing light with a larger effective optical window, overcomes the limitations between bandwidth, dark current, and QE for a given detector area. A typical device fabricated here features a high responsivity of 0.6 A/W, a low dark current of 138 nA, a high 3 dB bandwidth of 21.4 GHz, and a large effective optical aperture of 54 µm. Moreover, all fabricated devices have passed the reliability tests in accordance with the standards in the document Telecordia GR-468-CORE. Moreover, the demonstrated devices here can reach a 3 dB bandwidth of >35 GHz when the PD diameter is scaled down to 20 µm while maintaining the same large effective optical aperture, which may be applied for 400 Gbps pulse-amplitude modulation four-level (PAM4) application using a four-channel photo-receiver. All these results show great promise in utilizing the demonstrated Ge PD integrated polymer MLs for future high-speed optical communication systems.

## Figures and Tables

**Figure 1 sensors-24-04221-f001:**
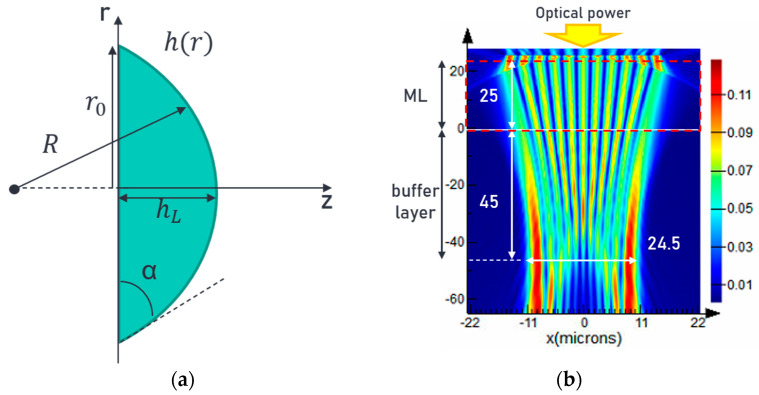
(**a**) The schematic of plano-convex ML showing the size of the ML *r*_0_, the radius of curvature *R*, the vertex height *h_L_*, and the contact angle *α*. (**b**) The FDTD simulation result showing the input light to the ML that is 30 µm in size focused from the MMF output light by a custom Rx lens module connected to a 50/125 µm MMF at 850 nm. The high-order mode profile of the MMF in the simulation is chosen so that the numerical aperture of the input light to the ML matches the experimental data. The ML is 20 µm thick (between 20 and 0 µm) and the buffer layer is 45 µm thick (between at 0 and −45 µm).

**Figure 2 sensors-24-04221-f002:**
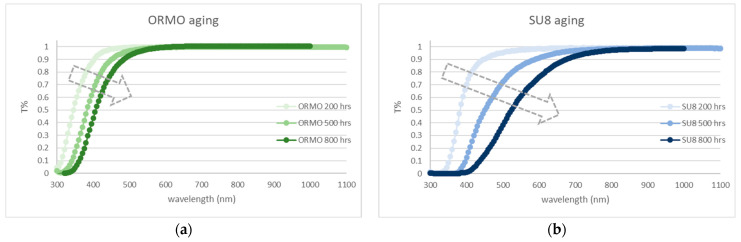
Transmission spectra of (**a**) ORMO and (**b**) SU8 derived from UV-VIS spectroscopy, after 200 h, 500 h, and 800 h of thermal treatment at 125 °C.

**Figure 3 sensors-24-04221-f003:**
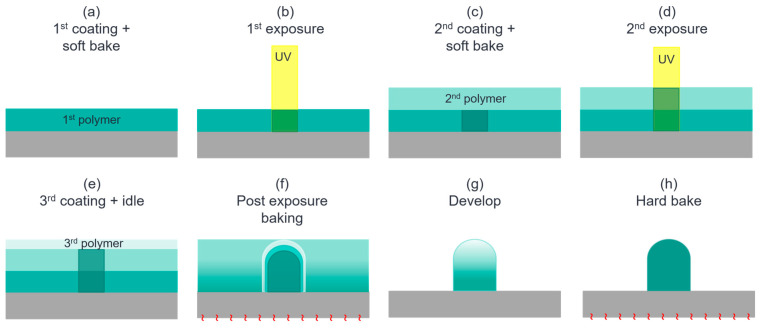
(**a**) The 1st spin-coating and soft bake to derive a uniform SU8 layer of 20 µm thickness. (**b**) The 1st UV exposure to define the pedestal dimensions. (**c**) The 2nd spin-coating and soft bake to place a uniform SU8 layer of 30 µm thickness over the 1st SU8 layer. (**d**) The 2nd UV exposure to define the ML dimensions and generate photon-acids in the 2nd SU8 layer. (**e**) The 3rd spin-coating of an SU8 layer, which is then left idle for about 1 hour to control the photon-acid diffusion. (**f**) PEB and (**g**) follow-up development performed to remove the extra polymer without a crosslink, forming the shape of the lens. (**h**) Hard bake to generate higher degrees of crosslink and dehydration.

**Figure 4 sensors-24-04221-f004:**
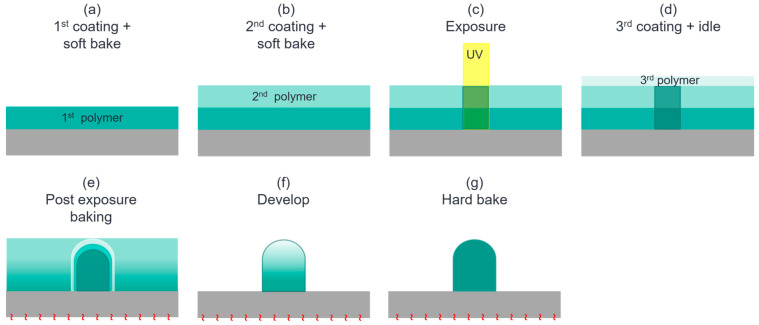
(**a**) The 1st spin-coating and soft bake to derive a uniform SU8 film of 20 µm thickness. (**b**) The 2nd spin-coating and soft bake to place a uniform SU8 layer of 30 µm thickness over the 1st SU8 layer. (**c**) UV exposure to define the ML and pedestal dimensions and generate photo-acids in the 1st and the 2nd SU8 layers. (**d**) The 3rd spin-coating of an SU8 layer, which is left idled for about 1 hour to control the photo-acid diffusion. (**e**) PEB and (**f**) follow-up development performed to remove the extra polymer without a crosslink, forming the shape of the lens. (**g**) Hard bake to generate higher degrees of crosslink and dehydration.

**Figure 5 sensors-24-04221-f005:**
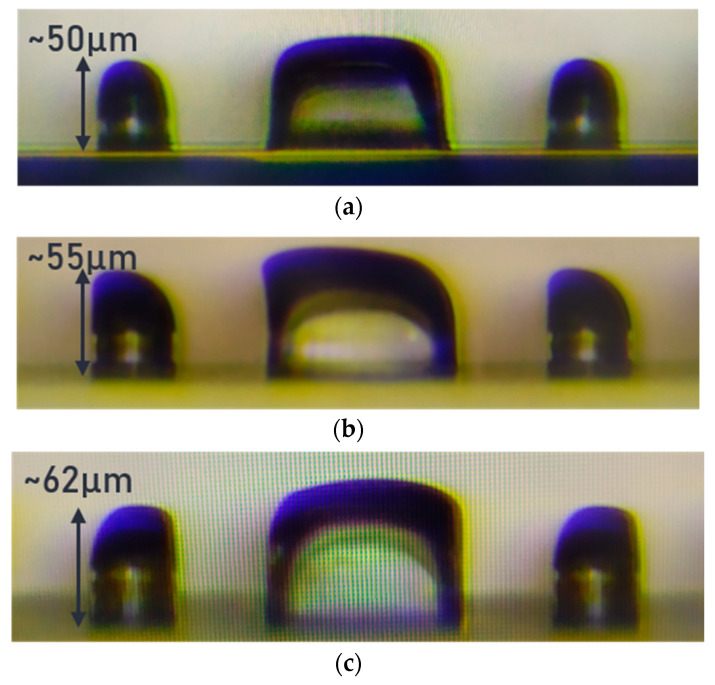
Two MLs on pedestals (leftmost and rightmost objects) formed by using Method 1 with a UV dosage of (**a**) 100 mJ/mm^2^ (**b**), 140 mJ/mm^2^, and (**c**) 160 mJ/mm^2^.

**Figure 6 sensors-24-04221-f006:**
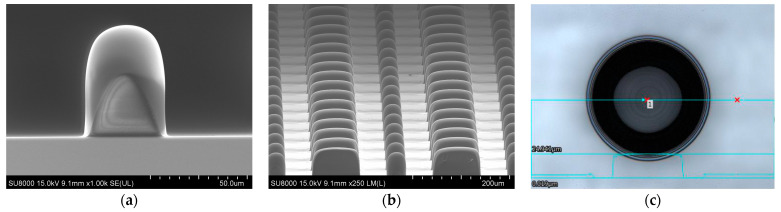
(**a**) SEM image of an ML on a pedestal with a 54 µm diameter fabricated by Method 2. (**b**) SEM image of ML-on-pedestal arrays, separated by spacer arrays, fabricated by Method 2. (**c**) Three-dimensional profile of fabricated ML using Method 2 for checking ML sphericity.

**Figure 7 sensors-24-04221-f007:**
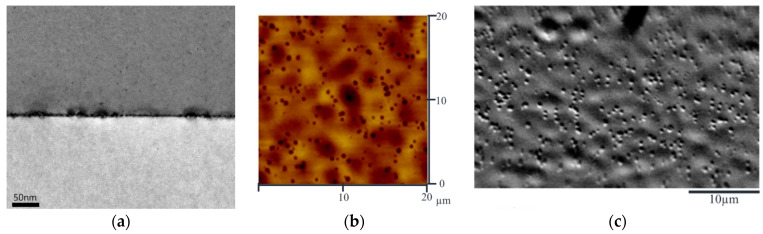
(**a**) Cross-sectional TEM image of the Ge film deposited on a Si substrate using RPCVD. (**b**) Top-view AFM image obtained by the Ge etch pit method. (**c**) Top-view SEM image obtained by the Ge etch pit method.

**Figure 8 sensors-24-04221-f008:**
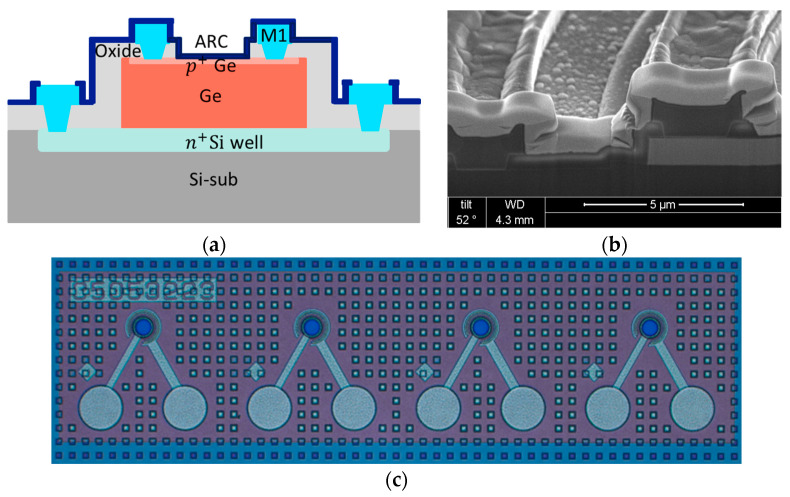
(**a**) Schematic plot of the Ge p-i-n PD. (**b**) Cross-sectional SEM image of the fabricated Ge p-i-n PD showing a via on top of Si and a via on top of Ge. (**c**) Top-view OM image of the fabricated four-channel Ge p-i-n PD array.

**Figure 9 sensors-24-04221-f009:**
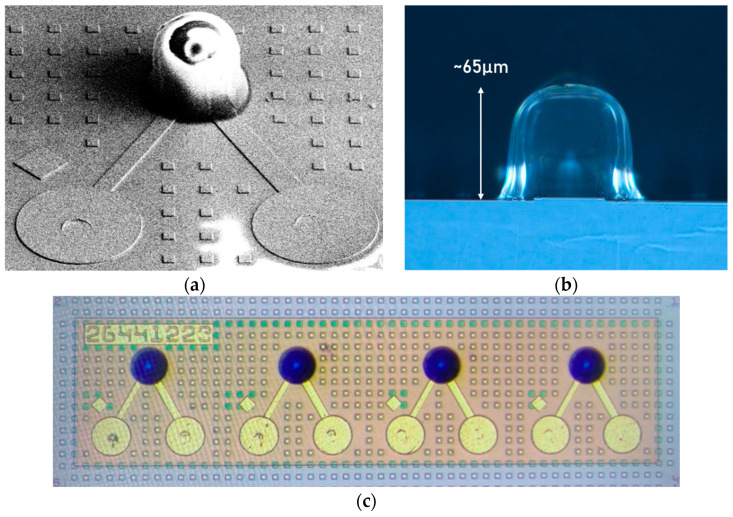
(**a**) A 45-degree view SEM image of an SU8 ML integrated on a Ge p-i-n PD. (**b**) Side-view OM image of the SU8 ML fabricated on a Ge p-i-n PD. (**c**) Top-view OM image of the four-channel Ge p-i-n PD array with integrated SU8 MLs.

**Figure 10 sensors-24-04221-f010:**
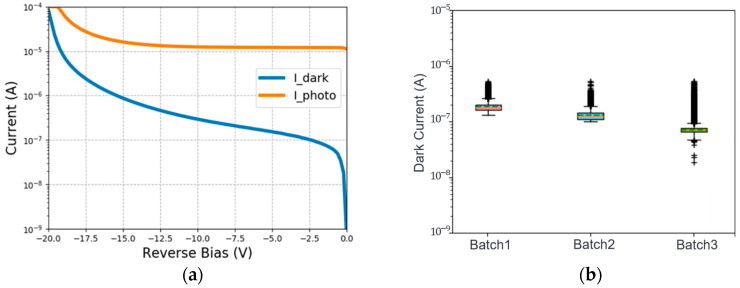
(**a**) Typical dark and photo I–V curves of the fabricated 29 µm diameter device. (**b**) Statistics of the dark currents at −3 V from three batches of the fabricated Ge p-i-n PDs.

**Figure 11 sensors-24-04221-f011:**
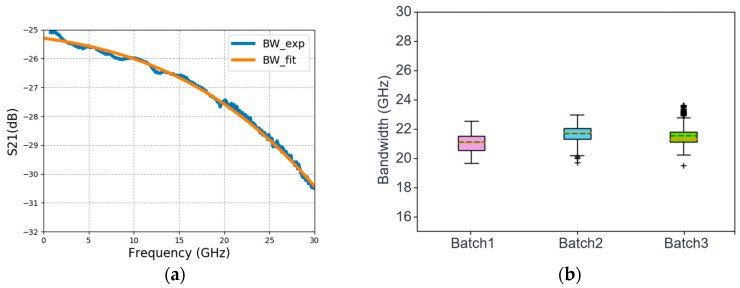
(**a**) Measured S21 frequency response. (**b**) Statistics of the 3 dB bandwidth from three batches of the fabricated Ge p-i-n PDs.

**Figure 12 sensors-24-04221-f012:**
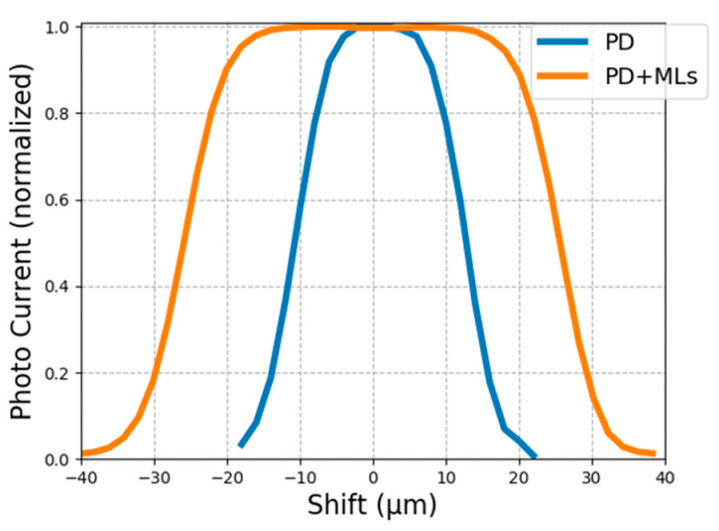
Photocurrent vs. displacement of the focuser center relative to the PD center.

**Figure 13 sensors-24-04221-f013:**
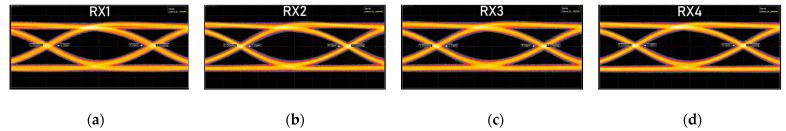
Eye diagram measured from the COB module for (**a**) the 1st channel, (**b**) the 2nd channel, (**c**) the 3rd channel, and (**d**) the 4th channel.

**Figure 14 sensors-24-04221-f014:**
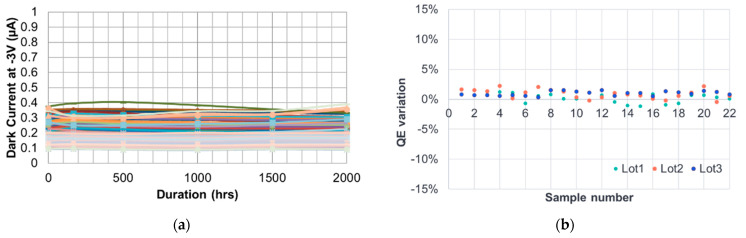
(**a**) HTOL reliability test at 175 °C showing dark current vs. duration up to 2000 h. (**b**) HTS reliability test at 135 °C showing QE variation for all samples after 2000 h.

**Table 1 sensors-24-04221-t001:** Comparison of vertical Ge p-i-n PD.

Ref.	Year	Wavelength (nm)	Ge THK (nm)	Ge Diameter (µm)	Optical Aperture (µm)	Responsivity (A/W)	QE (%)	Dark Current Density (mA/cm^2^)	Bandwidth (GHz)	Reverse Bias (V)	QE-BW Product (GHz)
[27]	2005	1310	2350	300	300	0.6	56.79	12 (@ −1 V)	-	-	-
		1550				0.52	41.6				
		1620				0.1	7.65				
[9]	2005	850	300	10	10	0.16	23	100 (@ −1 V)	38.9	−2	8.95
		1298				0.17	16				6.22
		1552				0.04	3				1.17
[8]	2009	1550	330	10	10	0.05	4	600 (@ −1 V)	39	−2	1.56
[15]	2010	1310	800	30	30	0.65	61.53	14.8 (@ −1 V)	12.6	−3	7.75
		1550				0.31	24.8				3.12
[14]	2020	1550	350	14	14	0.62	49.6	33 (@ −1 V)	33	−3	16.37
This work		850	1000	29	54	0.6	87.53	20 (@ −3 V)	21.4	−3	18.73

## Data Availability

The original contributions presented in the study are included in the article, further inquiries can be directed to the corresponding author.

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
