# Peer review of "Normal-Incidence Germanium Photodetectors Integrated with Polymer Microlenses for Optical Fiber Communication Applications"

_sensors, 2024, doi:10.3390/s24134221_

Round 1

Reviewer 1 Report

Comments and Suggestions for Authors

The authors presented a photon-acid diffusion method to integrate polymer microlenses on a 4-channel, high-speed photon receiver. It could absorb light through  an effective beam and overcome the limitations between bandwidth and dark current etc. The experimental results showed that the proposed scheme has a wide bandwidth, large effective optical aperture, and good robutness.

The research work was well organized and the experimental results sound interesting to the related researchers.

Author Response

(Please find the attachment.)

Reviewer 2 Report

Comments and Suggestions for Authors

Please check the attachment!

Reviewer 3 Report

Comments and Suggestions for Authors

This paper demonstrates an optimal method to fabricate polymer-based microlenses (MLs) through photon-acid diffusion. It is shown experimentally that MLs with effective optical window size as large as 54 µm can be mass produced and pass the related reliability qualifications. Such a ML allows the maintenance of high external responsivity through low coupling loss and high alignment tolerance when interfacing a photodiode. The integration of a Ge PD with the fabricated large polymer ML is demonstrated. The optical properties of the polymer ML and the Ge PD are designed through running commercial finite-difference time domain simulation software. The designed device is then fabricated and duplicated to form a 1D array on 8-inch Si wafers. The dark current, responsivity, 3dB bandwidth, and effective aperture size of the integrated device are characterized through either wafer-level or die-level electro-optical testing. Based on these results, selected devices are packaged and gone through eye diagram as well as sensitivity measurements. Moreover, reliability tests based on GR-468 reliability assurance standard are performed.

Some shortcomings of the paper are the following:

1. (Line 91): “…devices having a 3dB bandwidth of > 35 GHz to [25-27].” The sentence is not completed.

2. (Lines 159, 163, 216, 238, 271): Subscripts must be corrected for mm2, SiO2.

Comments on the Quality of English Language

There are some English language mistakes. For example:

(Line 2): “…Photodetector Integrate with…”

(Line 244): “…substrat”

(Line 376): “…sensitivities define by optical modulation amplitude…”

Reviewer 4 Report

Comments and Suggestions for Authors

The study describes the result of large-scale work to create a modern, competitive, 4-channel, high-speed photo-receiver consisting of germanium p-i-n photodiodes. The conclusions and numerical estimates presented in the study are confirmed by a sufficient number of experiments. The main focus of the study is on a novel photon-acid diffusion method to integrate polymer microlenses of the photo-receiver. There are minor shortcomings in the work, mainly related to the presentation of the results obtained.

Сomments:

1) Line 93-95. You need to remove the following text "This section may be divided by subheadings. It should provide a concise and precise description of the experimental results, their interpretation, as well as the experimental conclusions that can be drawn.".

2) Figure 1b. For clarity, in the figure you need to highlight and label the area corresponding to ML. For clarity, you need to indicate the direction of light propagation.

3) Figures 5, 8b and 9b. You need to specify the scale.

4) Line 217-226. The authors write: "Still, Method 1 cannot produce satisfactory ML shape when its diameter is roughly larger than 45 μm diameter. … On the other hand, ML on pedestal fabricated by Method 2 is shown in Fig. 6. … In Fig. 6(a), the scanning electron microscope (SEM) of a nearly perfect spherical ML on pedestal with a diameter of 54 μm is 224 shown…". But Figures 5 and 6 are not sufficient confirmation of these conclusions. It is necessary to provide in the text of the article estimates of deviation from sphericity when using Method 1 and Method 2.

5) Line 319-321.  The authors write: "Further measurements show that there is almost no observable difference between the dark currents measured from the Ge p-i-n PDs with and without the polymer MLs". If possible, this conclusion should be confirmed in Figure 10a (for example, provide graphs obtained with and without the polymer MLs).

6) Figures 10b and 11b should be improved. It's better to change the scale along the vertical axis (there's too much empty space in the Figures right now). The text of the article should briefly explain the graphic designation of statistical parameters (candlestick chart) used in Figures 10b and 11b. In Figures 10b and 11b, the graphs for batch 3 are too shifted along the horizontal axis (correct this).

7) Line 357-359.  The authors write: "Note that such sweepings are done in both x and y directions to examine the symmetry of the integrated ML, and we find no particular difference between the 2 directions implying good circular symmetry of the integrated ML". If possible, this conclusion should be confirmed in Figure 12 (for example, the graph corresponding to the shift in y can be indicated by a dotted line).
